# The Preservative Action of Protein Hydrolysates from Legume Seed Waste on Fresh Meat Steak at 4 °C: Limiting Unwanted Microbial and Chemical Fluctuations

**DOI:** 10.3390/polym14153188

**Published:** 2022-08-04

**Authors:** Eman T. Abou Sayed-Ahmed, Karima Bel Hadj Salah, Rasha M. El-Mekkawy, Nourhan A. Rabie, Mada F. Ashkan, Soha A. Alamoudi, Mohammed H. Alruhaili, Soad K. Al Jaouni, Mohammed S. Almuhayawi, Samy Selim, Ahmed M. Saad, Mohammad Namir

**Affiliations:** 1Department of Food Science, Faculty of Agriculture, Zagazig University, Zagazig 44511, Egypt; 2Biological Sciences Department, College of Science & Arts, King Abdulaziz University, Rabigh 21911, Saudi Arabia; 3Laboratory of Transmissible Diseases and Biologically Active Substances, Faculty of Pharmacy, University of Monastir, Monastir 5089, Tunisia; 4Department of Botany and Microbiology, Faculty of Science, Zagazig University, Zagazig 44511, Egypt; 5Medical Microbiology and Parasitology Department, Faculty of Medicine, King AbdulAziz University, Jeddah 21589, Saudi Arabia; 6Department of Hematology/Oncology, Yousef Abdulatif Jameel Scientific Chair of Prophetic Medicine Application, Faculty of Medicine, King Abdulaziz University, Jeddah 21589, Saudi Arabia; 7Department of Clinical Laboratory Sciences, College of Applied Medical Sciences, Jouf University, Sakaka 72388, Saudi Arabia; 8Biochemistry Department, Faculty of Agriculture, Zagazig University, Zagazig 44511, Egypt

**Keywords:** legume, wastes, enzymatic hydrolysis, protein, antioxidant, antimicrobial, buffalo meat steak, cold storage

## Abstract

Valorizing agricultural wastes to preserve food or to produce functional food is a general trend regarding the global food shortage. Therefore, natural preservatives were developed from the seed waste of the cluster bean and the common bean to extend the shelf life of fresh buffalo meat steak and boost its quality via immersion in high-solubility peptides, cluster bean protein hydrolysate (CBH), and kidney bean protein hydrolysate (RCH). The CBH and the RCH were successfully obtained after 60 min of pepsin hydrolysis with a hydrolysis degree of 27–30%. The SDS-PAGE electropherogram showed that at 60 min of pepsin hydrolysis, the CBH bands disappeared, and RCH (11–48 kD bands) nearly disappeared, assuring the high solubility of the obtained hydrolysates. The CBH and the RCH have considerable antioxidant activity compared to ascorbic acid, antimicrobial activity against tested microorganisms compared to antibiotics, and significant functional properties. The CBH and the RCH (500 µg/mL) successfully scavenged 93 or 89% of DPPH radicals. During the 30-day cold storage (4 °C), the quality of treated and untreated fresh meat steaks was monitored. Protein hydrolysates (500 g/g) inhibited lipid oxidation by 130–153% compared to the control and nisin and eliminated 31–55% of the bacterial load. The CBH and the RCH (500 µg/g) significantly enhanced meat redness (*a** values). The protein maintained 80–90% of the steak’s flavor and color (*p* < 0.05). In addition, it increased the juiciness of the steak. CBH and RCH are ways to valorize wastes that can be safely incorporated into novel foods.

## 1. Introduction

A short life distinguishes fresh products as they are an excellent medium for microbial growth and prone to lipid oxidation. The increase in microbial load and oxidation rate in food causes considerable economic losses and ethical problems worldwide. The deterioration in color and sensory traits of fresh meat is an obvious consequence of these problems [1]. Cutting fresh meat in steak form can effectively and efficiently ease the incorporation of additives that can enhance storage stability.

The Food and Agriculture Organization (FAO) revealed that food waste and food loss occur when “the quantity or quality of food decreases along the food supply chain” [2].

Food-spoiling microorganisms are the leading cause of food waste and loss. Approximately 20% of global meat production is lost due to microbial contamination. Controlling the population of microorganisms in animal products is one of the most important ways to reduce food waste [3]. In theory, increasing the levels of preservatives in food could be a solution to this problem. However, consumers dislike chemical preservatives that extend food shelf life [4]. As a result, there is a demand for natural alternatives to chemical preservatives.

Antimicrobial peptides (AMPs) are compounds that represent the first line of defense in plants, animals, and microbes against pathogenic microorganisms [5].

Biologically active peptides derived from vegetable wastes, especially legumes by enzymatic hydrolysis, are promising alternatives to chemical preservatives as they can extend the shelf life of food by inhibiting microbial contamination and food components’ oxidation. They are also biocompatible and biodegradable in vivo [6].

Enzymatic hydrolysis increased the human diet’s long-term bioavailability of amino acids, besides enhancing their absorbance through the small intestine [7]. Additionally, the protein hydrolysates have various activities. Recently, chickpea protein hydrolysate by chymotrypsin was isolated as an antimicrobial peptide against various foodborne pathogens, and it can be used as a food preservative [8]. A 2 h Flavourzyme-gram bean hydrolysate is characterized by a distinct structure and significant antioxidant activity [9]. Therefore, the application of these enzymatic hydrolysates has a significant impact on food formulation. Many trials are proceeding to incorporate various protein hydrolysates as potential natural preservatives into meat products and juices [10,11,12].

The antimicrobial mechanism of AMPs is briefed on reacting with certain parts of the bacterial membrane, such as anionic phospholipids and lipopolysaccharides, which break down the membrane and kill the bacteria [13]. The reaction may depend on peptides’ hydrophobicity by binding the hydrophobic groups in the membrane. Additionally, in ionic/electrostatic interactions, the peptides are deposited on the bilayer surface, causing cellular membrane flux and disintegration [14,15]. Furthermore, amphipathicity, when peptides contain hydrophobic and hydrophilic residues, both previous mechanisms may be functioning. In addition, peptide length plays a critical role; short peptide has a good amphipathic structure with powerful antimicrobial activity [16,17].

Legume wastes, i.e., broken seeds, which consumers do not prefer, are precious sources of nutrients, especially protein [18]. However, many of these biomaterials do not get used and wind up in municipal landfills, causing major environmental problems and negative economic impacts. Therefore, managing massive amounts of various degradable materials is challenging [19].

Several studies highlighted the extraction of protein from food wastes (FW). They concerned the extraction of protein from FW to strengthen the concept of recycling and to utilize valuable extracted protein from FW as an equally valuable recycled ingredient and product to induce sustainability. So, the extraction processes often target protein yield, and they must be environmentally greener. Enzymatic hydrolysis in our study achieved this equation by excluding harsh chemicals. The extraction steps of protein from food waste are of prime importance to maximize protein yield and quality, where polysaccharide removal may affect protein solubility. Solubility is a marker of protein extractability. Tabal et al. [20] provided a new protein hydrolysate from pigeon pea milling waste (26% protein) for use in food formulation. In addition, Tassoni et al. [21] highlighted the use of pea, bean, and chickpea agro-industrial wastes in preparing protein for the formulation of feed, food, cosmetic, and packaging products.

However, there are no studies on valorizing the unwanted cluster bean (52% protein) and common red bean (25% protein) seeds to produce novel, eco-friendly protein hydrolysates and incorporate them in food formulation.

Cluster bean or guar (*Cyamopsis tetragonoloba*) is a valuable legume. Its seeds contain 52.6% protein and high mineral and vitamin content, mainly Fe and Vitamin C [22]. Green or dried seeds have numerous medicinal and industrial applications for humans and animals [23]. Additionally, red kidney beans (*Phaseolus vulgaris* L.) are an excellent source of protein (20–40%), consumed as an inexpensive protein source in many developing countries [24].

In this study, we explored novel pepsin protein hydrolysates from cluster bean and common red bean wastes (CBH and RCH) which have not been used in previous studies—that were chemically characterized by SDS-PAGE. Functional properties, e.g., the hydrolysate’s solubility, water-holding capacity, and oil holding capacity were evaluated and associated with their activities (antioxidant and antimicrobial). Buffalo meat steak was covered with CBH and RCH hydrolysates and stored for 30 days under cold conditions, while continuously monitoring the chemical and microbial changes.

## 2. Materials and Methods

### 2.1. Protein Hydrolysates Isolation

The seed wastes of the cluster (*Cyamopsis tetragonoloba*) and kidney beans (*Phaseolus vulgaris* L.) were finely ground by a Moulinex blender (FP823125, France). The powder was homogenized in hexane (1:3, w:v). The protein was isolated from defatted powder as per Millan-Linares et al., [25]

The cluster bean and the common bean protein isolates were blended with pepsin (0.5%), homogenized in acidic phosphate buffer pH 2, and put in a heat bath (37 °C) for intervals of 0, 30, 45, and 60 min. The enzyme was inhibited at 90 °C for 15 min. The solution was centrifuged at (14,000× *g*, 5 min) to obtain CBH and RCH, lyophilized, and kept for further analysis [11,26].

### 2.2. The Degree of Hydrolysis (DH)

The %DH of cluster bean and common bean protein isolates after 0, 30, 45, and 60 min were determined by Holye and Merrltt [27]. A total of 100 µL of Trichloroacetic acid (10%) was added to 100 µL of protein isolates, then centrifuged under cooling at 12,298× *g* for 10 min. The total nitrogen in the supernatant protein and TCA was measured by the Kjeldahl method [28].

### 2.3. Characterization of Cluster Bean and Common Bean Hydrolysates

#### 2.3.1. SDS-PAGE

After pepsin hydrolysis for 30 and 60 min, the protein hydrolysates were separated by discontinuous SDS-PAGE (18%) (Arabian Group for Integrated Technologies “Agitech”, New Cairo, Egypt). The buffer system was (Tris HCl, pH 6.8 for staking gel, and Tris HCl, pH 8.8 for resolving gel) following Laemmli [29]. A total of 5 µL of protein in loading sample buffer was loaded in each well. A 5 to 245 kDa Tris-Glycine marker was embedded to configure the detected bands in electropherogram, which was stained by Coomassie brilliant blue.

#### 2.3.2. Physicochemical Analysis of Hydrolysates

The suspensions of protein hydrolysates and seed residues were served for the following analysis. Moisture content was determined using AOAC method 925.10 [28]. The protein content was evaluated by elemental microanalysis as % nitrogen content × 6.25, using a NB9830 full automatic kjeldahl protein analyzer by AOAC method 920.87 [28]. The ash content was evaluated using the direct ignition method (550 °C for 25 h), AOAC method 923.03 [28]. The fat content was determined using AOAC method 945.16 [28]. Carbohydrate content was determined by difference. Carbohydrate was calculated by subtracting the sum percentage of moisture, protein, fat, ash, crude and dietary fiber. The protein hydrolysates used in the following analysis were obtained after 60 min of pepsin hydrolysis.

#### 2.3.3. Functional Properties of Protein Hydrolysates

##### Solubility

The solubility of CBH and RCH (60 min of pepsin hydrolysis) was estimated at different pH (2–10) according to the method described in Saad et al. [12], with some modification. The CBH and the RCH (0.1 mL) were suspended in 25 mL of distilled water, stirred for 45 min at 45 °C, while adjusting pH, and then centrifuged under cooling at (5000× *g*, 10 min) to estimate the total nitrogen in the protein supernatant using the Kjeldahl method [28], which was then applied in Equation (2).
(1)$$Solubility\;\left(\%\right)=\frac{Amount\;of\;protein\;in\;the\;supernantent}{Amount\;of\;protein\;in\;sample}\times 100$$

##### Water Absorption Capacity

In weighted test tubes, 100 mg of CBH and RCH were stirred with 10 mL of sterilized distilled water for 30 min. The tubes were centrifuged at 6000× *g* for 30 min. The supernatant was suspended, then the tubes remained tilted at 45° for 30 min until the surface water was broken, then reweighted [30].
(2)$$water\;absorbing\;capacity=\frac{absorbed\;water\;\left(g\right)}{sample\;weight\;\left(g\right)}$$

##### Oil Absorption Capacity

In weighted test tubes, 500 mg of CBH and RCH were homogenized in 10 mL of oil for 30 min. The tubes were centrifuged for 30 min at 6000× *g*, the supernatant was discarded, and the tubes were left upside down for 30 min to remove surface oil before being reweighted [30].
(3)$$oil\;absorbing\;capacity=\frac{absorbed\;oil\;\left(g\right)}{sample\;weight\;\left(g\right)}$$

### 2.4. Total Phenolic Compounds (TPC) in the Hydrolysates

Total phenolic content was determined by the Folin–Ciocalteu reagent according to the method of Müller et al. [31]. In brief, in a 96-well microplate, 20 microliters of CBH and RCH obtained after 60 min pepsin hydrolysis were added to 100 microliters of Folin–Ciocalteu reagent and 75 microliters of Na_2_CO_3_ solution (7.5%), then incubated for 60 min in the dark. The absorbance was read at 765 nm by a microplate reader. The total phenolic content was expressed as g gallic acid equivalent/mL of protein by applying the following equation: y = 0.004x + 0.1257.

### 2.5. Biological Activity of Pepsin Protein Hydrolysates

#### 2.5.1. Antioxidant Activity

The scavenging potential of the DPPH free radical was determined as per Gali and Bedjou [32]. In brief, 160 µL of DPPH solution was added to 40 µL of CBH and RCH levels (50, 100, 200, 300, and 500 µg/mL), and ascorbic acid (500 µg/mL) was used as a reference. After 30 min of incubation in a dark place at 25 °C, the absorbance was read at 517 nm. The ability of protein hydrolysates to scavenge the DPPH radical was applied in this Equation (5):(4)$$Radical\;scavenging\;activity\;\left(\%\right)=\frac{Abs.\;control-Abs.\;sample}{Abs.control}\times 100$$

#### 2.5.2. Antimicrobial Activity

The microbial strains, *Bacillus cereus* (BC), *Listeria monocytogenes* (LM), *Staphylococcus aureus* (SA), *Escherichia coli* (EC), *Pseudomonas aeruginosa* (PA), *Salmonella typhi* (ST), *Aspergillus niger* (AN), *Aspergillus flavus* (AF), *Candida gelbeta* (CG), *Candida tropicalis* (CT), and *Candida albicans* (CA) were taken from the microbial culture collection (MIRCN) in the Faculty of Agriculture, Ain Shams University, Egypt. The bacterial inoculum was prepared at 1 × 10^8^ CFU/mL, and the fungal inoculum was prepared at 1 × 10^5^ CFU/mL.

##### Antibacterial

The disc diffusion method was used to assess antibacterial activity. Discs (6 mm) were saturated with CBH and RCH at concentrations of (25, 50, 100, 200, 400, and 500 µg/mL) and placed on the surface of bacterial strains-inoculated Muller Hinton agar (MHA) plates and incubated at 37 °C for a day. The positive control was penicillin (500 µg/mL), and water was used as negative control. The diameters of the inhibition zones (mm) were calculated [12,33].

The minimum inhibitory concentration (MIC) and minimum bactericidal concentration (MBC) were estimated [12,33]. For MIC, 50 µL of CBH or RCH concentrations were added to Muller Hinton broth (MHB) tubes inoculated with bacterial stains. The tubes were incubated at 37 °C for 1 day before the turbidity was measured at 600 nm. For MBC, 100 µL of MIC tubes were spread on new MHA plates that were incubated at 37 °C, and bacterial population was observed after 24 h.

##### Antifungal

The disc diffusion technique was used to assess the antifungal activity of CBH and RCH [34,35]. The *Candida* and fungal inoculum were prepared in Sabouraud dextrose (SD) broth at a concentration of 10^5^ CFU/mL. The prepared inoculum was spread over Sabouraud dextrose Agar (SDA) plates. CBH and RCH saturated 6 mm discs with different concentrations were placed on SDA plates, while clotrimazole (500 µg/mL) saturated discs were used as a positive control. The SDA plates were incubated for 2 days at 37 °C and 5 days at 30 °C for *Candida* and fungi, respectively. The distances of the inhibition zones (mm) were estimated. The MIC and MFC were determined with microdilution broth and MFC with spreading plates as mentioned in antibacterial activity, considering the incubation conditions of *Candida* and fungi.

### 2.6. Preservation of Fresh Meat Steak

Two factors were studied in this experiment: 1. Concentrations of CBH and RCH (0, 100, 250, and 500 µg/g meat); and 2. Storage time: 0, 10, 20, and 30 days.

Meat steaks were immersed in 100 mL of CBH and RCH suspension concentrations (0, 100, 250, and 500 µg/mL) for 24 h at 4 °C before being packed in polyethylene bags and stored in refrigeration at 4 °C for analysis.

Meat steaks were divided into 8 equal proportions and mixed with different concentrations of CBH, RCH, and nisin according to the following formulations: Using the immersion method, CBH 0, 100, 250, and 500 µg/g were T1-T4; RCH 0, 100, 250, and 500 µg/g were T5-T7, and nisin (500 µg/g) was T8, respectively. The samples were packaged and stored in a refrigerator at 4 °C. A total of 8 random samples were taken for analysis during the storage period (0, 10, 20, and 30 days).

#### 2.6.1. Physicochemical Analysis of Meat Sample

##### pH and Glycogen Content Estimation

The pH of steak samples was measured by Ibrahim et al. [36]. Steak samples (10 g) were homogenized in 100 mL of distilled water for 1 min. The pH was then measured by a pH meter. Glycogen content in meat samples was measured according to Dreiling et al. [37].

##### Metmyoglobin (MetMb) Analysis

The MetMb content in meat samples was estimated by Krzywicki [38]. The treated and untreated steak samples (1 g) were blended for 10 s in a magnetic stirrer with 10 mL of ice-cold 0.04 M phosphate buffer at pH 6.8 before being centrifuged at 3000× *g* for 30 min at 4 °C. The supernatant was further clarified by filtration through Whatman No. 1 filter paper. The absorbance of the filtrate was measured at 525, 572, and 700 nm using a UV-VIS spectrophotometer (Shimadzu, Nakagyo-ku, Kyoto, Japan). The %Met-Mb concentration was estimated by Equation (6) [38].
(5)$$MetMb\;\left(\%\right)=\left[1.395-\frac{A572-A700}{A525-A700}\right]\times 100$$

##### The Percentage of Inhibition in Lipid Peroxidation (LPI)

The Witte et al. [39] method was used to estimate the percentage of LPI. The meat steak samples were suspended in cold 50 mM phosphate buffer (pH 7) and centrifuged at high speed (14,000× *g*, 30 min, 4 °C). The obtained supernatant (100 µL) was mixed with barbituric acid (2 mL) and boiled for 30 min before cooling. A spectrophotometer (Shimadzu, Nakagyo-ku, Kyoto, Japan) was used to measure the sample absorbance at 530 nm. The percentage of LPI was calculated in Equation (6):(6)$$Lipid\;oxidation\;inhibition\;\left(\%\right)=\left[1-\frac{Sample\;absorbance}{control\;absorbance}\right]\times 100$$

#### 2.6.2. Sensorial Properties and Color Measurement

The steak samples were evaluated for sensory characteristics, including color, flavor and aroma, tenderness, juiciness, and overall acceptability. Steak samples (after a 30-day storage period) were cooked in an oven at 176 °C for 8.5 min until the internal temperature reached 70 °C, then served warm at 60 °C to eight trained panelists [40]. Steak samples from different treatments were randomized and evaluated within the session. Water was mounted after each sample assessment. Panelists rated each sample attribute using a 9-point hedonic scale. The higher score values indicate a greater preference.

Concerning the meat color, the Hunter color analyzer (Hunter Lab color Flex EZ, USA) was used to measure the color parameters (*L**, *a**, and *b**) of meat steak samples [41].

#### 2.6.3. Microbiological Analysis

The microbial load of buffalo meat steak was calculated as per Saad et al. [42]; 10 g of meat steak samples were homogenized with 90 mL sterilized buffer peptone water for 10 min to prepare a 10^−1^ concentration; decimal dilutions were prepared up to 10^−6^; in one-use Petri-dishes, 1 mL of each dilution was added, followed by the appropriate media; on plate count agar (PCA), the total bacterial count (TBC) was determined after a 24-h incubation period at 30 °C. Additionally, after a 10-day incubation period at 7 °C, psychrophilic bacteria counts (PBC) were counted at PCA [43]. The microbial counts were transformed to logarithms (CFU/g).

### 2.7. Statistics

All experiments were done three times. The average of the replicated data was analyzed by One Way ANOVA at a probability level of 5%, followed by an LSD test to define the significant differences between means using Microsoft Excel (v. 2019).

## 3. Results and Discussion

### 3.1. The Approximate Composition of Protein Isolates and Hydrolysates

Table 1 presents the proximate analysis of the seed wastes and their protein hydrolysates of cluster and kidney beans. Cluster bean seed wastes and their protein hydrolysate contain high protein, i.e., 57.2 and 93.2%, respectively, compared to 23.35 and 88.9% in red kidney beans. Furthermore, red common bean seeds have the highest carbohydrate content, recording 64.1%. Ash content was higher in cluster bean hydrolysates, reaching 7.3%. In a previous study, the approximate composition of black beans was carbohydrates (71.4%), protein (23.1%), ash (4.3%), and fat (1.2%). However, the protein content in its protein isolates increased by 253% compared to black bean seed, while carbohydrates decreased by 11.4% [44].

Generally, ash content increased with pepsin hydrolysis in CBH rather than RCH. The results indicated that CBH protein content was significantly increased by 63%. However, the increase was 282% in RCH because of the increase of 70% in ash content in CBH. A prior study found a comparable decrease in protein level in Alcalase-black kidney bean protein hydrolysate, probably because of the increase in ash content, which may be due to the addition of NaOH to maintain the pH during hydrolysis [44]. In Alcalase-hydrolyzed chickpea protein, a comparable reduction in protein content was observed, along with an increase in ash content [45].

### 3.2. Physiochemical Characterization of the Hydrolysates

#### 3.2.1. SDS-PAGE Electropherogram

Figure 1 shows an SDS-PAGE electropherogram of the obtained CBH and RCH after pepsin hydrolysis for 30 min and 60 min. In total, 10 bands in the range of 17–100 kD in RCH were detected in lane 1. However, 5 bands of 17–48 kD in CBH were detected in lane 3. These 15 bands correspond to storage proteins, where 47–75 kD refers to vicilin (7S), and 40–48 kD refers to phaseolin (8S), following Los et al. [46], the 8S and the 7S bands in RCH after 30 min hydrolysis are more intense than those in CBH. After 60 min of pepsin hydrolysis, 7S bands (50–63 kD) still existed in lane 2 (RCH), but no bands were detected in lane 4, indicating the complete hydrolysis of CBH, which agreed with Saad et al. [11], who found the total disappearance of white kidney bean protein bands after 6 h of pepsin hydrolysis.

#### 3.2.2. The Degree of Hydrolysis (DH)

Figure 2 depicts the DH (%) of cluster and red common bean protein isolates after 60 min of pepsin hydrolysis at 37 °C. In terms of hydrolysis time dependence, the DH was significantly increased. After 60 min of pepsin hydrolysis, the maximum DH of CBH and RCH was recorded, reaching 30 and 27%, respectively. The CBH had a high DH by a relative rise of nearly 11% over the RCH. Saad et al. [11] found the degree of hydrolysis was 33.3% for white kidney bean protein after hydrolysis by pepsin 1% for 6 h. Additionally, pepsin black bean protein was hydrolyzed by 27% with 2 h of pepsin hydrolysis [47]. In addition, when Bumrungsart and Duangmal [9] used Flavourzyme^®^ (6%) for 6 h to hydrolyze black gram bean protein isolate, they obtained a high DH (75%).

#### 3.2.3. The pH-Protein Solubility

Incorporating functional protein into food formulation is depends on its solubility, influencing protein foaming and emulsifying properties [48]. The CBH and the RCH had isoelectric pH (lowest solubility) of 4–6 similar to the intact mother protein (Table 2). Protein solubility increased significantly (*p* < 0.05) when the pH shifted away from the isoelectric point. After 60 min of pepsin hydrolyzed, the solubility of CBH and RCH were 80 and 75% at pH 3, which increased to 100 and 90% at pH 11, respectively. The order of solubility level is CBH > RCH, which is related to the degree of hydrolysis and means that the solubility is improved. The fact that the basic side has a higher solubility than the acidic side is consistent with Los et al. [46], who found the solubility of carioca bean and soybean protein hydrolysate at pH 3.0 was 35.13%; but, when pH was raised to 10, the hydrolysates dissolved. Furthermore, the solubility of papain-kidney bean protein hydrolysate at pH 10 was 78% [49].

#### 3.2.4. Functional Properties

Table 2 shows the water-holding capacity (WHC) and oil-absorbing capacity (OAC) of CBH and RCH after 0-, 30-, and 60-min pepsin-hydrolysis. The CBH has the highest WHC, with a solubility improvement of 22% over the RCH. After 60 min of pepsin hydrolysis, the OAC of CBH was increased by 11% compared to RCH. In addition, Eckert et al. [50] observed an increase from 6.12 to 8.21% in OAC of faba bean protein after pepsin hydrolysis. Enzymatic hydrolysis is commonly used to improve the functional properties of proteins. Incorporating plant proteins instead of animal proteins in food formulation is a new trend in the food industry [51,52]. Plant proteins’ nutritional value and functional properties are critical [53]. High-solubility proteins improve the technical qualities of fortified foods and are required in many food applications [54].

#### 3.2.5. Total Phenolic Content

The total phenolic content of different protein hydrolysates is presented in Figure 3A. The content of polyphenols grew in a concentration-dependent manner. The CBH (500 µg/mL) had higher phenolic compound values with 75.4 mg GAE/g, which increased 10% above RCH. Dark (44.3 mg gallic acid/g) and red (38.89 mg GAE/g) bean protein hydrolysates have comparable phenolic contents [55]. Protein–phenolic interactions may affect protein physicochemical properties; peptide activity may be increased by hindering specific residues of amino acids, thereby increasing the polyphenol absorption and activity [56]. Enzymatic hydrolysis increased the total polyphenols by breaking down the protein–polyphenol complexes and releasing some polyphenols entrapped in the peptide fragments [55,57].

#### 3.2.6. Biological Activity

##### Antioxidant Activity

The DPPH-scavenging ability of protein hydrolysates is presented in Figure 3B. Most of the DPPH radical (93%) was scavenged by CBH (500 µg/mL). The antioxidant activity of protein hydrolysates depends on phenolic compounds in a concentration-dependent manner. The high polyphenol content in CBH and RCH is accountable for the higher scavenging activity of these hydrolysates. The scavenging power of pepsin-kidney bean hydrolysate was 85%, and for papain-kidney bean hydrolysate, it was 89% [11,49]. The scavenging power of CBH was stronger than intact protein or ascorbic acid (500 µg/mL), which is used commercially in the food industry [55]. The mode of action of antioxidant hydrolysates depends on making free radicals stable by donating electrons or transferring protons from aromatic amino acids in hydrolysates. Furthermore, the acidic amino acids may stabilize free radicals by sharing a proton with the NH_2_ and COOH residues [58]. Antioxidant peptides are critical in the food industry because they prevent the oxidation of protein, lipid, and nucleic acid, ensuring product quality [59].

##### Antimicrobial Activity

Table 3 illustrates the inhibition zone (DIZ, mm) distances of the mentioned meat-borne bacteria and fungi when exposed to different concentrations of CBH and RCH (25, 50, 100, 200, 400, and 500 µg/mL). CBH (25–500 µg/mL) caused the biggest DIZs of 13–35 mm, followed by RCH (12–32 mm). The bacteria most vulnerable to the protein hydrolysates were *S. aureus* and *E. coli*, demonstrating 26 and 35 mm DIZ, respectively. However, *P. aeruginosa* and *L. monocytogenes* were the most resistant isolates to the hydrolysates, with an estimated DIZ of 28 and 23 mm, respectively.

The resistant Gram-negative bacteria have a lower DIZ than the Gram-positive. The unique membrane structure of Gram-negative bacteria is attributable to their resistance to antibacterial drugs through their lipopolysaccharide layer and certain enzymes in the periplasmic area [60,61].

The lowest MIC values (20–45 µg/mL) were recorded in CBH-treated MHA plates, with a relative decrease of 40% compared to RCH (Table 4). The lowest CBH concentration that could kill the tested bacteria was 45–80 µg/mL, compared to 65–145 µg/mL for RCH at higher doses. There are similar values of DIZ in red kidney bean hydrolysate; 19.23 mm and 20.26 mm against *P. aeruginosa* and *E. coli*, respectively. Additionally, the MIC of papain-kidney bean protein hydrolysate was 70–90 µg/mL [49].

The type of bacteria determines the antibacterial effect of peptides. The peptides may bind electrostatically to the bacterial membranes, and the membrane rigidity and cell components could be impaired [62].

Table 3 shows the antifungal activity of CBH and RCH. The DIZs of tested hydrolysates (25, 50, 100, 200, 300, 400, and 500 µg/mL) were in the range of (8–35 mm) against tested *Candida* and fungi. The results showed no significant differences between the antifungal activities of hydrolysates on fungi or *Candida*. The fungi vulnerable to CBH (500 µg/mL) were **Candida tropicalis* (CT)* and **Candida albicans* (CA)*, with DIZs of 32–35 mm. However, the resistant fungi, *C. gleberta,* and *A. niger* recorded DIZs of approximately 24 and 27 mm, respectively.

Table 4 shows that the MIC and MFC of CBH are lower than those of RCH. The CBH suppressed fungal growth at MICs of 35–40 mm and MFCs of 75–90 mm; MIC and MFC against fungi and MIC and MBC against bacteria; the fungi had higher values (Table 4), indicating that hydrolysates have more powerful antibacterial than antifungal activity. The proposed mechanisms suggest that hydrolysates are more suited for bacteria than fungi, which have a more complex structure than bacteria. Heymich et al. [63] stated that chickpea protein hydrolysate exhibited potent antibacterial activity against *Escherichia coli* and *Bacillus subtilis* with minimum inhibitory concentrations of 31.3–62.5 µM. Additionally, it displayed antifungal activity with minimum inhibitory concentrations of 125–500 µM against *Saccharomyces cerevisiae* and *Zygosaccharomyces bailii*. In addition, the purified peptide from the seeds of the cowpea plant displayed higher MIC values against fungal spoilers, with MICs of 50 μg/mL against *F. culmorum* and > 500 μg/mL against *Penicillium expansum* [64].

### 3.3. Fresh Meat Steak Preservation by Cluster and Common Bean Protein Hydrolysates

#### 3.3.1. Physiochemical Alternation during Cold Storage

The physiochemical alternations of treated and untreated fresh meat steaks during storage of 0, 10, 20, and 30 days at 4 °C are presented in Table 5. The studied parameters included tracking the changes in pH, metmyoglobin (%), and lipid oxidation inhibition (%) in response to CBH and RCH addition (0, 100, 250, and 500 µg/g) with a preference for CBH. All parameters’ values considerably increased in storage period dependence, but lipid oxidation inhibition (%) significantly decreased in concentration dependence. The pH values of the control fresh steak increased by 52% from the start of storage until day 30 of cold storage.

The pH in supplemented meat samples with RCH and CBH (500 µg/g) significantly decreased by 30–33% compared to the control at the end of the storage period. The gradual increase in the pH value with storage time is probably a result of food spoilage microorganisms that can hydrolyze the proteins into NH_3_ [65]. The addition of CBH and RCH (500 µg/g) to the stored meat significantly reduced this value in a concentration-dependent manner. This action appears to be due to the previously demonstrated antimicrobial activity of the protein hydrolysates. Similar results were observed by Saad et al. [11] and Saad et al. [12] on using common bean protein hydrolysates in preserving minced beef or chicken meat.

The estimated glycogen content in meat samples is 1.1 µmol/g in the control sample, 0.8 µmol/g in RCH-supplemented meat, 0.5 µmol/g in CBH-supplemented meat, and 0.6 µmol/g in nisin-supplemented meat. The low glycogen content didn’t affect the pH value of meat and that agrees with England et al. [66].

The Met-myoglobin levels in the untreated samples dramatically increased by 416% at the end of storage, exceeding the acceptable level in meat (40%). The treated meat samples with CBH and RCH (500 µg/g) were characterized by low metmyoglobin levels of 21–26%, which decreased by 96–142% compared to the untreated samples. The considerable decrease (142%) was achieved by CBH (500 µg/g). Increasing metmyoglobin levels in stored meat significantly affect the meat color because of the oxidation of myoglobin and the auto-oxidation of protein in the meat [67].

The antioxidant activity of added hydrolysates (Figure 3) is probably responsible for the met-myoglobin reduction. A decrease in met-myoglobin was observed when treating beef burgers with cinnamon, rosemary, and thyme extracts [68]. Lipid oxidation is one of the main factors that affect meat quality by producing hydroperoxides and aldehydes [1]. Therefore, adding RCH and CBH (500 µg/g) to meat steak dramatically reduces lipid oxidation by approximately 130–153% compared to the control.

The antioxidant activity of CBH and RCH is probably responsible for scavenging the radicals produced by lipid oxidation [1]. Moreover, Pineul et al. [69] found inhibition of lipid oxidation by 82% in zebrafish meat treated with red bean hydrolysate. Additionally, Aslam et al. [70] discovered that storing supplemented chicken breast with fish protein hydrolysate delayed lipid oxidation and other undesirable changes.

#### 3.3.2. The Fluctuation in Color Parameters and Sensory Properties of Stored Buffalo Steak for 30 Days at 4 °C

Table 6 and Figure 4A present color parameter fluctuations in fresh meat steak treated or untreated with CBH and RCH during a storage period of 30 days. The lightness values (*L**) of meat steak decreased with CBH and RCH (500 µg/g) in concentration dependence by 6–10% compared to the control. The dark color of hydrolysates may cause this decrease. However, *a** value increased by 4–6% compared to the control. Additionally, blueness values increased from 3.08 to −2.50 in CBH-treated samples and from 3.08 to −0.5 in RCH-treated steak. In general, color components deteriorated at the end of storage, but protein hydrolysate inclusion significantly preserved approximately 80–90% of color attributes based on the protein hydrolysates compared to nisin and the control.

No studies shed light on the alternations in color components and sensory characteristics of fresh meat steak enriched with CBH. However, the obtained results were in agreement with Arshad et al. [71], who studied the beneficial effects of bioactive peptides on the oxidative stability and functional properties of beef nuggets supplemented with milk casein protein hydrolysates at levels of 0, 2, 4, 6, and 8% during cold storage of 15 days.

The Hunter color test also revealed a significant difference between groups, where lightness values of beef nuggets were decreased in 8% supplemented samples and during the storage period. In addition, yellowness decreased, but the redness of beef nuggets increased. On the other hand, the addition of CBH and RCH considerably kept the meat quality parameters at high levels (Figure 4B–F). The highest values of tenderness and juiciness, 8.6 and 8.8, were observed in CBH-supplemented steak because of the highest scores in water-holding capacity in CBH (Table 2).

CBH enhanced cooked steak flavor and taste, followed by RCH supplementation. All quality parameters decreased at the end of cold storage, but the precious roles of CBH and RCH significantly reduced the unwanted changes by 55–60% (Figure 4B–F) compared to nisin and the control.

The above-mentioned sensorial scores affected the meat’s overall acceptability scores, where CBH-supplemented fresh steak was safe for approximately 28–29 days of cold storage, while the RCH-supplemented steak (500 µg/g) was safe for 20 days.

Similar results have been noticed in the sensorial traits of raw buffalo meat supplemented with pea, and red kidney bean hydrolysates were maintained and were highly acceptable compared to the control concerning shelf-life [72].

#### 3.3.3. Microbial Alternation in Fresh Meat Steak during 30 Days Storing at 4 °C

Figure 5 shows a considerable (*p* < 0.05) increase in bacterial load during the storage period at refrigeration. At the end of the storage period, the total bacterial count declined in supplemented fresh buffalo steak with RCH, CBH, and nisin at a high concentration (500 µg/g). As a result, the total bacterial count in supplemented samples was reduced by 31–55% compared to controls.

The antibacterial activity of CBH could extend the secure cold storage of fresh meat for approximately 20–30 days, when kept refrigerated. Saad et al. [11] found that found that adding kidney bean protein hydrolysate to minced beef lowered the microbial load by 22%. Additionally, Sharma et al. [73] discovered that adding natural preservatives to chicken sausages, such as a herbal blend, considerably reduced the microbial population. The allowable bacterial count in fresh buffalo meat must be less than 1 × 10^6^ CFU/g, according to Egyptian Standards No. 4334 [74] and the International Commission on Microbiological Specification (ICMS, 1982).

## 4. Conclusions

Food waste harms the environment, but legumes are rich in nutrients, so maximizing these wastes is critical. Enzymatic hydrolysis and microbial fermentation can quickly and precisely produce bioactive peptides. CBH and RCH can be exploited as possible antioxidants and efficient antibacterial in food systems. They can extend the cold lifetime of preserved fresh steak to 28–29 days while sustaining acceptable sensory qualities.

## Figures and Tables

**Figure 1 polymers-14-03188-f001:**
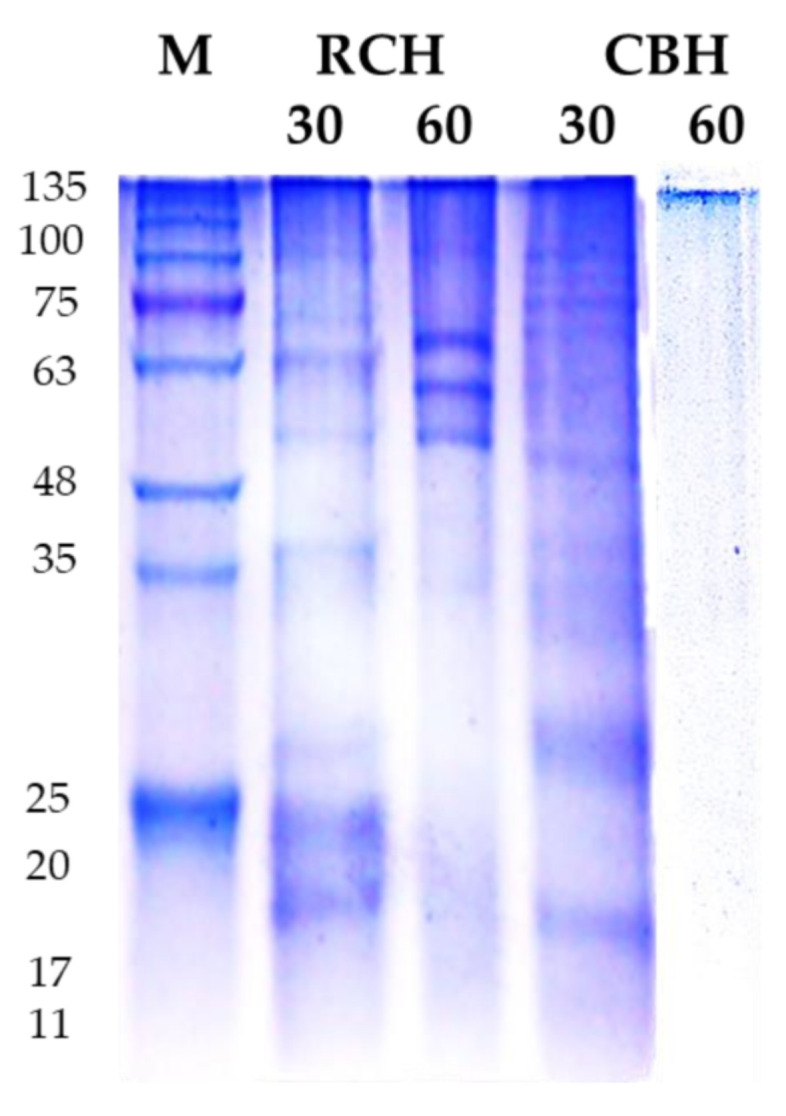
SDS electropherogram (18%) of RCH and CBH isolated from cluster and red kidney bean seeds wastes. Lane 1, M, molecular marker (Tris-Glycine SDS-PAGE, 4–20%). Lanes 2 and 4 represented protein bands of protein isolates after 30 min pepsin hydrolysis; lanes 3 and 5, represented 60-min pepsin hydrolysis at 37 °C of protein isolates. Buffer system, discontinuous SDS-PAGE buffer system (Tris HCl, pH 6.8 for staking gel, and Tris HCl, pH 8.8 for resolving gel). 5 µL of protein in loading sample buffer was loaded in each well.

**Figure 2 polymers-14-03188-f002:**
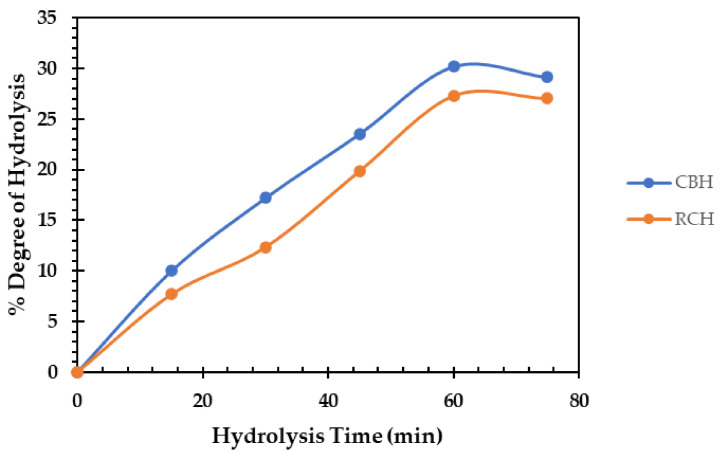
Degree of hydrolysis (DH) of cluster bean hydrolysate (CBH) and red bean protein hydrolysates (RCH) (60 min pepsin hydrolysis at 37 °C).

**Figure 3 polymers-14-03188-f003:**
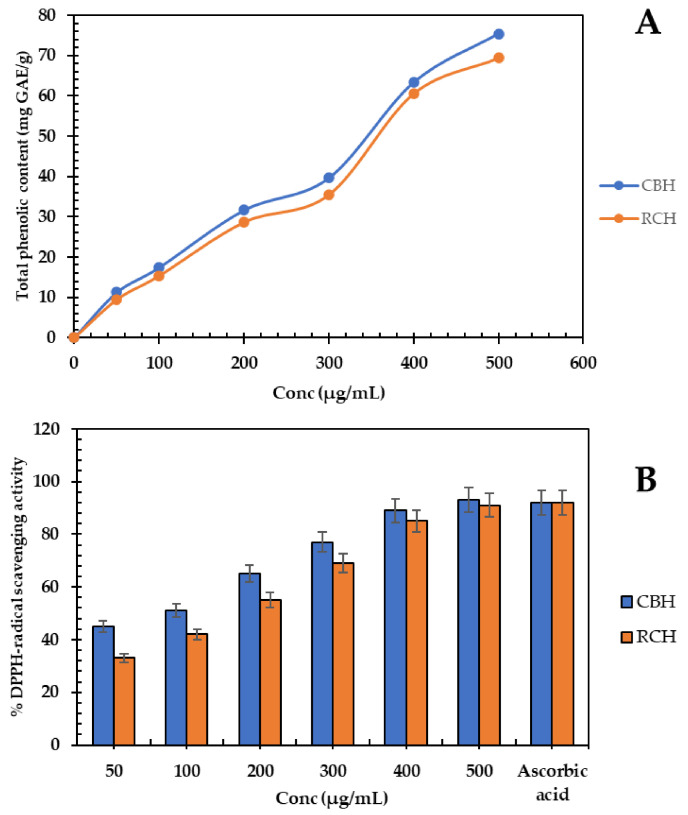
(**A**), Total phenolic content (mg gallic acid equivalent/g) of CBH and RCH concentrations (50–500 µg/mL). (**B**), DPPH˙ scavenging activity of CBH and RCH concentrations compared to ascorbic acid (500 µg/mL).

**Figure 4 polymers-14-03188-f004:**
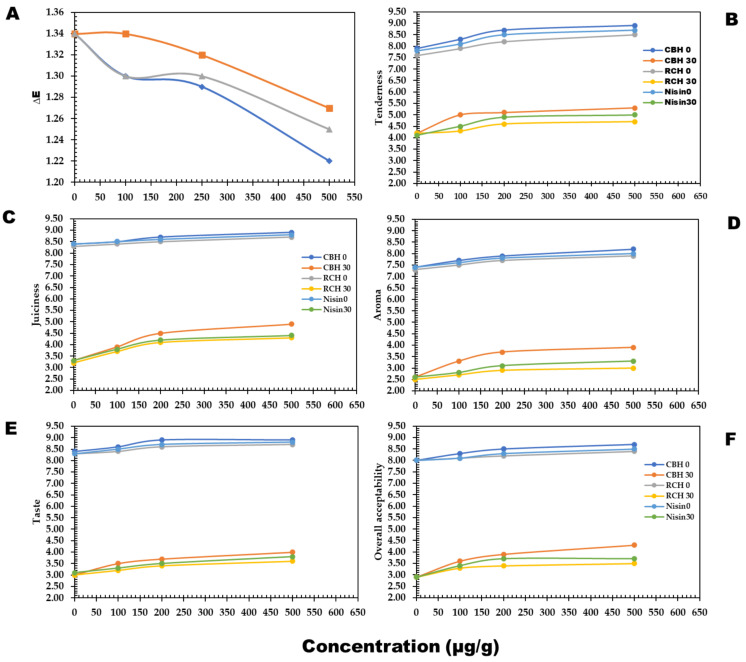
The changes in sensorial quality of buffalo meat steak supplemented with CBH and RCH (100, 250, and 500 µg/g) and nisin 500 µg/g persevered at 4 °C during 0–30 days storage. The sensorial parameters; Color (**A**), Tenderness (**B**), Juiciness (**C**), Aroma (**D**), Taste (**E**), and Overall Acceptability (**F**).

**Figure 5 polymers-14-03188-f005:**
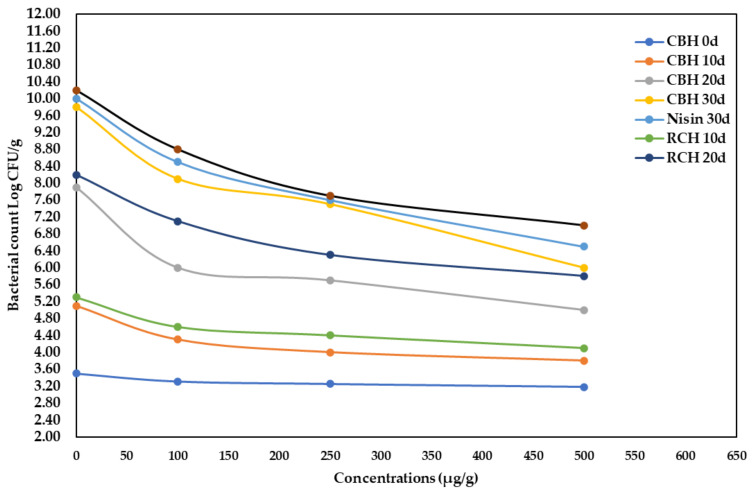
The alteration of total bacterial count Log CFU/g of stored buffalo steaks at 4 °C during 0- and 30-days storage period, as supplemented with CBH, RCH (100, 250, and 500 µg/g), and Nisin at 500 µg/g.

**Table 1 polymers-14-03188-t001:** Approximate analysis of seed wastes and protein hydrolysate of cluster and red kidney beans.

Proximate Composition (%)	Beans Seed Wastes			
	Cluster		Red Kidney	
Material Status	Seed	Hydrolysate	Seed	Hydrolysate
Protein	57.2 c	93.2 a	23.35 d	88.9 b
Carbohydrates	30.8 b	0.13 d	64.1 a	4.6 c
Fat	2.25 a	-	2.3 a	-
Ash	4.3 c	7.3 a	4.19 c	6.5 b
Moisture	6.1 a	-	6.3 a	-

The lowercase letters next to values means in the same row indicate significant differences at *p* ≤ 0.05. Protein hydrolysate is obtained after 60 min of hydrolysis with pepsin at 37 °C.

**Table 2 polymers-14-03188-t002:** Functional properties (water-absorbing capacity, oil-absorbing capacity, and solubility) of CBH and RCH were obtained after 60 min of pepsin hydrolysis at 37 °C.

Protein Hydrolysate	HT (min)	Functional Properties						
		WAC (g/g)	OAC (g/g)	Solubility (%)				
				pH 3	pH 5	pH 7	pH 9	pH 11
CBH	0	6.3 e	6.9 e	20 b	5 d	11 c	22 b	35 a
	30	8.7 c	10.22 c	50 ab	12 d	24 c	41 b	64 a
	60	11.6 a	13.66 a	80 ab	20 d	55 c	72 b	100 a
RCH	0	6.1 e	6.5 e	14 bc	5 d	9 c	19 b	29 a
	30	7.6 d	8.3 d	35 b	10 d	22 c	38 b	55 a
	60	9.4 b	11.96 b	75 ab	14 d	41 c	60 b	90 a

Means with different lowercase letters in the same raw indicate significant differences between hydrolysates solubility at *p* ≤ 0.05. WAC, water-absorbing capacity; HT, hydrolysis time; OAC, oil-absorbing capacity. Means with different lowercase letters in the same column indicate significant differences between WAC and OAC values.

**Table 3 polymers-14-03188-t003:** The diameters of inhibition zones (mm) of CBH and CBH at different concentrations of 25–500 µg/mL against Gram-positive, Gram-negative bacteria and fungi.

Bacteria	CBH						RCH					
	25	50	100	200	300	400	25	50	100	200	300	400
G+												
*B. cereus* (BC)	-	14 ab	20 b	22 b	25 b	27 b	-	13 ab	18 b	21 b	23 b	26 b
*L. monocytogenes* (LM)	-	13 b	18 c	20 c	24 c	26 c	-	12 b	17 c	20 c	22 c	25 c
*S. aureus* (SA)	-	15 a	22 a	25 a	28 a	32 a	-	14 a	21 a	23 a	26 a	30 a
G-												
*E. coli* (EC)	-	12 a	17 a	19 a	22 a	24 a	-	-	15 a	18 a	20 a	23 a
*P. aeruginosa* (PA)	-	9 c	13 c	15 b	19 c	21 c	-	-	11 c	14 c	17 c	19 c
*S. typhi* (ST)	-	10 b	15 b	18 ab	20 b	22 b	-	-	13 b	16 b	18 b	20 b
Fungi												
*A. niger* (AN)	-	8 b	12 d	13 d	17 d	22 d	-	-	9 c	11 b	15 d	20 d
*A. flavus* (AF)	-	9 ab	13 c	14 c	19 c	23 cd	-	-	11 ab	13 ab	17 bc	21 c
*C. gelbeta* (CG)	-	8 b	11 bc	14 c	17 b	24 c	-	-	9 c	11 b	15 d	21 c
*C. tropicalis* (CT)	-	10 a	14 a	18 a	21 a	29 a	-	-	12 a	14 a	19 a	26 a
*C. albicans* (CA)	-	9 ab	12 b	16 b	20 ab	27 b	-	-	10 b	13 ab	18 b	24 b

The lowercase letters next to values means indicate significant differences *p* < 0.05.

**Table 4 polymers-14-03188-t004:** The lowest concentration (µg/mL) of CBH, RCH, and antibiotic, inhibiting microbial strains, (MIC), and the lowest concentration (µg/mL) killing bacterial strains, (MBC), and fungal strains, (MFC).

**Bacterial Strain**	**CBH (µg/mL)**		**RCH (µg/mL)**		**Antibiotic * (µg/mL)**	
	**MIC**	**MBC**	**MIC**	**MBC**	**MIC**	**MBC**
BC	30 d	60 e	35 e	70 e	25 d	50 d
LM	35 c	65 d	40 d	75 d	30 c	60 c
SA	30 d	50 f	35 e	65 f	20 e	45 e
EC	40 b	70 c	65 c	125 c	35 b	70 b
PA	45 a	90 a	80 a	155 a	40 a	80 a
ST	40 b	75 b	70 b	130 b	35 b	70 b
**Fungal strain**	**MIC**	**MFC**	**MIC**	**MFC**	**MIC**	**MFC**
AN	40 b	80 b	80 b	165 a	40 a	85 a
AF	30 c	75 c	70 d	140 c	35 b	70 b
CG	45 a	85 a	85 a	150 b	40 a	85 a
CT	30 c	70 d	65 e	125 e	30 c	65 c
CA	30 c	75 c	75 c	135 d	35 b	70 b

Different lowercase letters next to means indicate significant differences at probability level 5%. * Penicillin with Bacteria, clotrimazole with fungi. Minimum inhibitory concentration (MIC), minimum bactericidal concentration (MBC), minimum fungicidal concentration (MFC).

**Table 5 polymers-14-03188-t005:** Physicochemical fluctuation in raw buffalo steak supplemented with cluster bean and red common bean protein hydrolysate (CBH and RCH) compared to nisin during 0–30 days cold storage at 4 °C.

Sample	Conc (µg/g)	Cold storage Time (days)											
		pH				Metmyoglobin (%)				Lipid Oxidation Inhibition (%)			
		0	10	20	30	0	10	20	30	0	10	20	30
Control	0.0	5.8 a	6.33 a	7.50 a	8.80 a	10.00 a	27.00 a	43.00 a	51.60 a	38 d	30 d	22 d	13 g
CBH	100	5.7 b	5.89 c	6.25 c	7.00 c	9.1 ab	19.00 c	21.00 b	30.45 d	37 ab	33 c	30 c	20 e
	250	5.65 b	5.74 c	6.20 c	6.55 d	8.2 bc	14.50 cd	16.00 c	26.00 e	37 ab	35 b	33 ab	28 c
	500	5.49 c	5.67 c	5.88 d	6.14 e	7.67 c	11.99 d	15.00 c	21.65 f	38 a	37 a	34 a	33 a
RCH	100	5.89 a	6.00 b	6.54 b	7.35 b	9.8 a	21.30 b	25.90 b	36.00 b	36 b	31 d	28 d	17 f
	250	5.72 b	5.9 c	6.32 c	6.87 cd	8.7 b	16.55 c	23.10 b	32.50 c	36 b	34 c	31 bc	24 d
	500	5.56 c	5.79 c	6.25 c	6.44 d	7.9 c	14.33 cd	16.80 c	26.35 e	37 ab	36 ab	32 b	30 b
Nisin	500	5.51 c	5.70 c	6.12 c	6.20 e	7.7 c	12.54 d	15.56 c	24.66 ef	37 ab	36 ab	33 ab	32 ab

Means with different lowercase letters in the same column indicate significant differences.

**Table 6 polymers-14-03188-t006:** The alternation in color parameters in buffalo steak supplemented with CBH and RCH at graded concentrations (0, 100, 250, and 500µg/g) during 0–30 days cold storage.

Sample	Conc	Storage (Day)						
	(µg/g)	0	30	0	30	0	30	∆E
		*L**		*a**		*b**		
Control	0	49.55 a	48.66 a	14.00 c	13.70 c	4.00 a	3.80 a	1.34 a
CBH	100	48.70 b	47.92 b	15.10 a	14.80 b	1.90 c	1.10 c	1.29 b
	250	46.54 c	46.00 c	14.90 b	14.30 bc	−0.90 e	−1.80 ef	1.30 ab
	500	44.65 d	44.22 d	15.20 a	15.00 a	−1.50 f	−2.50 f	1.22 c
RCH	100	49.00 ab	48.12 ab	14.20 c	13.98 c	2.30 b	2.00 b	1.34 a
	250	48.22 bc	47.57 b	14.40 c	14.00 bc	0.50 d	0.20 d	1.32 ab
	500	46.33 c	45.88 cd	14.80 b	14.20 bc	−0.50 e	−1.10 e	1.27 b

Means with different lowercase letters in the same column indicate significant differences at *p* ≤ 0.05. Cont.: control. The Lightness (*L**) [(0–100) lightness to darkness], redness (*a**) [(− to +) redness to greenness], and *b** value reflected (+) yellowness to (−) blueness.

## Data Availability

Not applicable.

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
