# Peer review of "The Preservative Action of Protein Hydrolysates from Legume Seed Waste on Fresh Meat Steak at 4 °C: Limiting Unwanted Microbial and Chemical Fluctuations"

_polymers, 2022, doi:10.3390/polym14153188_

Round 1
Reviewer 1 Report
Summary:
The article describes the application of two different bean protein hydrolysates for preservation of fresh raw buffalo meat cut into steaks. Protein extracts of ground seed wastes from cluster and kidney beans were hydrolyzed with pepsin and used as preservative. The main contribution of this manuscript is that especially the cluster bean hydrolysate clearly increased the shelf life of the buffalo steaks in a concentration dependent manner, both in terms of microbial count, oxidation state, pH, as well as taste. Different experiments to analyze the pure peptide hydrolysates’ physiochemical properties and biological activities were conducted to try to analyze the background of the observed preservative effects. The authors showed that both bean hydrolysates had a concentration dependent effect in all conducted biological and physiochemical assays within the range of concentrations that was tested (0-500 µg/ml). In addition, the authors claim to predict the effect that higher than tested concentrations of the hydrolysates will have on meat preservation.
Comments
The corresponding author of the study has conducted extensive research on the preservative activities of different bean protein hydrolysates on foods and beverages. The experiments conducted in this study seem to be sound and support most of the claims made in the manuscript.
Major issues
Why or how the authors claim a predictive nature based on linear regressions on mostly non-linear relationships remains elusive. This predictive quality should either be removed or further explained.
Most of the experiments seem to lack a positive control, which could serve as a reference for readers to evaluate the activities observed. In the case of the antioxidant assay shown in figure 3B, the concentration of the control is lacking (ascorbic acid).
The use of references needs reviewing. In some cases, there seem to simply be the wrong references (r61 line 352), but in other cases the references seem to be out of context with the discussion (lines 348 and 372 r61) or wrongly/imprecisely cited (372 r60). In addition, the concept of antimicrobial activity of peptides in food preservation including the modes of action could be highlighted in more detail already as part of the introduction.
Some figure texts and/or axis descriptions are incomplete or inaccurate. Figure text 3 lacks information about the assay conducted. Figure text 4 lacks information about the measurement method and threshold for the definition of MIC. In the same figure the used bacterial strains are called isolates.
Figure 6 lacks information about the use of log decades on the y axis and the units on the x-axis.
Figure 1 lacks information about the amount of protein lysate added and information about the ladder as well as the PAA-concentration and buffer system used.
No information about the origin of the microbial test strains is given.
Minor issues:
Line 4: remove the rest of the title after Fluctuation (if not further elucidated in the manuscript)
Line 33: change to µg/ml
Line 36 the control and … the bacterial load
Line 45 change this sentence for example to: …medium for microbial growth and prone to lipid oxidation
Line 49 probably should read cutting instead of restricting fresh meat
Line 56 unclear what unwated seeds from consumers refers to, are they sent back by the consumers?
Line 72 maybe “in addition” instead of “also”
Line 75 unclear what “at long release” refers to as reference is not openly accessible
Line 76 intestine instead of intestines
Line 78 what is a significant structure activity?
Line 82 “we introduced novel pepsin hydrolysates” maybe change to “we explored novel pepsin hydrolysates”
Line 86 remove “The”
Line 98 the enzyme was denaturated at 90 degrees for how long?
Line 110 should give gel producer and/or concentration and buffer system
Line 149 Antimicrobial methods should contain strain origins and bacterial concentrations in assays
Line 175 packed
Line 180 7 of how many altogether?
Line 215 homogenized with what?
Line 233 “Alternatively” is probably not the right expression here
Line 235 approximate instead of proximate
Line 263 bands hardly visible on gel, minutes could be written on top of each lane, lane 1 has the marker lane 2 the first sample and so on
Line 269 change “was observed” to “is shown”
Line 288 logically improves what?
Line 312 remove “dramatically”
Line 331 ascorbic acid at which concentration?
Line 345f write Gram-positive/negative
Line 366ff review your description of the proposed models for antimicrobial peptide activity and mention amphipathicity
Line 376 What are bacteria more suited for than fungi? Isn’t it the hydrolysate, which is more suited for inhibition of bacteria than fungi? When it comes to structure, that depends what you are looking at.
Line 381 would be nice with a positive control and maybe consider showing the MIC as values in a table
Line 389 what do you mean with preference?
Line 394 how fresh was the meat? What about the glycogen contents effect on the pH?
Line 417 long list of names probably in addition to reference
Line 462 this figure and all the following show measuring concentrations of 200 µg/ml while you consistently claim 250 µg/ml in the figure texts and other texts
Line 479 why 28-29 days? Where is the data?
Line 482 replace alternation with alteration
Line 485ff conclusion seems too long
Author Response
Dear Reviewer
Thanks for the valuable comments that greatly enhanced the manuscript. Could you see the response letter below

Reviewer 2 Report
1. Abbreviation in the text should be defined. CBH and RCH should be defined more clearly.
2.There are so many reports on the bioactive peptides. How about the novelty of the manuscript? The author should explain why it is interesting to do the experiments they describe and especially what is new compared to these published papers. It might be highlighted that the new material could realize the high-value utilization of wastes or by-products. The incipit has to be supported with proper suitable literature references.
doi: 10.1016/j.foodhyd.2021.107180, 10.3390/polym11020238.
3.The development of bioactive substances based on natural ingredients capable of extending the food shelf life in a safe manner, especially antibacterial materials, should be mentioned.
doi: 10.1016/j.lwt.2021.111617,doi:10.3390/foods9040449, doi:10.1016/j.jfoodeng.2021.110697, doi:10.1016/j.foodhyd.2018.11.051
The extraction of bioactive substances from Legumes Seed should be introduced. Some very recent literatures could be used as examples.
doi: 10.3390/polym11050785
The Author could also cite other suitable literature.
4.Section 2 should be described with more details. For example, what was used as control for the Physicochemical analysis of meat sample? How about the concentration of ascorbic acid? How to proceed the Proximate analysis of seed wastes and protein hydrolysate?
5.It might be better to rewrite the Figure legends .
6. For most characterizations, the hydrolysis time of the hydrolyzate used was not clarified.
Author Response
Dear Reviewer,
Many thanks for your suggestions for enhancing the manuscript. We have replied to all suggestions in the response letter.

Round 2
Reviewer 1 Report
I am still sceptical concerning the predictive qualities of the study, concerning higher than tested concentrations.
Author Response
Response: We thank the reviewer for his efforts in reviewing the manuscript and improving its scientific quality.
Based on the valuable opinion of the reviewer in this round and the previous round, all the points related to the prediction quality were deleted, as the research topic depends on the valorization of legume residues in the production of protein hydrolysates with preservation effectiveness against microbes and oxidation.
We realized that the prediction point would not affect the main aim of the work, so we removed it.
Thanks again to the reviewer.

Reviewer 2 Report
The axes in the figures should have clear tick marks。
The introduction was wordy.
References seem to be too many for a research paper
Author Response
The axes in the figures should have clear tick marks
Response: Thanks for the reviewer; the tick marks were added to all figures
The introduction was wordy.
Response: Thanks for the reviewer; the introduction was reduced accordingly
References seem to be too many for a research paper
Response: Thanks to the reviewer; the references were reduced from 85 to 74
